# Preparation and Characterization of PHBV/PCL-Diol Blend Films

**DOI:** 10.3390/polym15244694

**Published:** 2023-12-13

**Authors:** Tamara Erceg, Sanja Rackov, Pal Terek, Branka Pilić

**Affiliations:** 1Faculty of Technology Novi Sad, University of Novi Sad, 21000 Novi Sad, Serbia; sanja.rackov@uns.ac.rs (S.R.); brapi@uns.ac.rs (B.P.); 2Faculty of Technical Sciences, University of Novi Sad, 21000 Novi Sad, Serbia; palterek@uns.ac.rs

**Keywords:** poly(3-hydroxybutyrate-co-3-hydroxyvalerate) (PHBV), poly(caprolactone diol) (PCL-diol), blend, biopolymers, film-forming ability

## Abstract

Biodegradable thin films based on poly(3-hydroxybutyrate-co-3-hydroxyvalerate) (PHBV) and poly(caprolactone diol) (PCL-diol) blend were developed using the solution casting method. PHBV is biodegradable, biocompatible, and produced naturally by bacterial activity, but its use is restricted by high crystallinity and low resistance to thermal degradation with melting temperatures close to degradation thus narrowing the processing window. Solution casting was chosen as a cost-effective method reducing energy consumption and avoiding thermal degradation during processing. The increase in PCL-diol in blend composition (40–60 wt%) enhances the film-forming ability of PHBV and the wettability along with the decrease in the roughness of the resulting materials as revealed by contact angle measurements, scanning electron microscopy (SEM), and atomic force microscopy (AFM). Optimal composition in terms of filmogenity and surface structure has been achieved by the addition of PCL-diol in the amount of 60 wt%. FTIR confirmed the expected chemical structures with no evidence of chemical interactions between the two polymers.

## 1. Introduction

The persistence of conventional plastic products opens up serious environmental concerns, directing tremendous societal attention and demand for novel bio-based and biodegradable solutions. Although the use of non-degradable plastics has proven to be reasonable due to economical and practical advantages including low cost, optical properties, low density, ease of processing and integration in production lines, and excellent resistance, once it reaches its useful life, it represents a significant source of waste that is not aligned with the worldwide strategies for energy and material recovery [1,2]. Key issues are related to their production pathways through extraction from non-renewable and limited petrochemical sources, their accumulation both in soil and water resulting from inadequate waste disposal, and/or a short end-of-life cycle. Although the majority of plastic waste is recyclable, most of it is mismanaged and disposed of in landfills or incinerated for practical and economic reasons. Consequently, considerable academic and industrial effort contributes to the global trend of developing a sustainable economy, which includes, inter alia, a systematic approach to production and extending the application of bioplastics [3,4,5]. In this sense, among the wide range of promising substitutes, poly(3-hydroxybutyrate-co-3-hydroxyvalerate) (PHBV) is distinguished by its fabrication route through bacterial fermentation from biomass. PHBV synthesis can be attained by a wide variety of microorganisms, such as Gram-negative (i.e., *Pseudomonas*, *Azobacter*) and Gram-positive ones (i.e., *Streptomyces*, *Rhodococcus*, *Bacillus*), and promoted by their unbalanced growth throughout the fermentation. PHBV and generally poly(hydroxyalkanoates) (PHAs) accumulation is a survival mechanism of bacterial cells and their response to controlled conditions (limited macro-elements and excess of carbon source) [6,7]. PHBV is a linear isotactic polyester composed of hydroxy acid units. It is compostable in industrial conditions and, in some cases, degradable under environmental conditions, which makes it even more attractive from a renewability perspective. Furthermore, its composition, chain length, and physicochemical properties can be tailored by variations in selected bacterial strains, nutrient sources, and applied conditions and as a result display a range of mechanical and thermal properties. Despite its substantial feature is total biodegradability, PHBV also exhibits resistance to hydrolysis and better resilience to UV degradation than polypropylene (PP) [8].

Although their high crystallinity degree provides good gas barrier performance, it is also the cause of low ductility and high brittleness, with the consequent low elongation at break. The main drawback that renders PHBV unsuitable to introduce on an industrial scale is the narrow processing window due to the proximity of the melting and degradation onset temperatures. High production costs, together with the above-mentioned technical disadvantages, limited the expansion of their utilization and compatibility with conventional thermal processing techniques [9,10]. Since PHBV is not yet competitive with conventional plastic, and in neat form, does not meet the high expectations as an engineering material in terms of mechanical properties, numerous scientific and industrial efforts are currently underway to ensure improved performance. To address these limitations and facilitate its processability, certain modification strategies have been established including co-polymerization, the use of plasticizers, fillers, and other modifiers, as well as blending with other polymers [11,12,13]. The possibility of performing these modifications makes PHBV suitable to be investigated for application in food packaging and biomedicine using different processing procedures such as extrusion, injection molding, 3D printing, electrospinning, and solvent casting. Applying the design of composite material, Gupta et al. have improved the tensile strength of PHBV by blending this biopolymer in a tween-screw extruder with polybutylene adipate-co-terephthalate (PBAT) and hemp powder, using a maleic anhydride-grafted PBAT as a compatibilizer [14]. Ioanna Kontogianni et al. have used the same processing method for the preparation of poly-L-lactic acid (PLLA), polycaprolactone (PCL), and poly(3-hydroxybutyrate-co-3-hydroxyvalerate) (PHBV) blend filaments enriched with nano-hydroxyapatite and strontium-substituted nano-hydroxyapatite intended for the preparation of scaffolds for the promotion of in vitro osteogenic activity [15]. In combination with akermanite, PHBV biopolymer has been proven to be a promising material for the design of scaffolds with potential applications in bone tissue engineering using the technique of selective laser sintering (SLS) [16]. Using injection molding, a group of authors has developed sustainable composites based on biobased poly (butylene succinate) and PHBV reinforced with talc and starch as potential alternatives to single-use plastic packaging. The addition of PHBV has led to an improvement in the barrier properties of poly(butylene succinate) [17]. 

Considering PHBV’s low thermal stability close to the melting point and therefore narrow operational window, electrohydrodynamic processing at room temperature by means of electrospinning and electrospraying has also gained considerable interest during the last few years [8,18]. Similarly, solvent-cast films based on PHBV blends are considered to be cost-effective and scalable, avoiding thermal processing [19]. Zhang and Mohanty reported a significant increase in flexibility and toughness achieved by the preparation of ternary blends of polylactide (PLA), PHBV, and poly(butylene succinate) (PBS), while the crystallization rates of PLA and PBS were enhanced by the presence of PHBV crystals acting as a nucleation agent [20]. Naseem and coauthors investigated the influence of polycaprolactone (PCL) and PHBV on PLLA thermal and mechanical properties intending to overcome the drawbacks of the individual polymers. The results show that the addition of PCL and PHBV in the concentrations of 5–10% to PLLA improves the average modulus of elasticity by up to 25%, average strength by up to 50%, and also the value of average elongation at break by 4000% [21]. Other authors have described the preparation of PHBV/PCL blends by means of solution casting and electrospinning, showing the superior tensile modulus and tensile strength of solvent-cast blends in comparison to electrospun ones. While PHBV/PCL obtained by solvent casting resulted in higher values for elongation at break but lower tensile strength than neat PHBV, for electrospun PHBV/PCL, these values decreased with the increase in the PCL content [19]. Different authors have reported the production of nanofibers of PHBV by electrospinning and the addition of polyethylene oxide (PEO) [22] and poly(3-hydroxybutyrate) (PHB) for the same intention: to modulate the mechanical characteristic of PHBV through blending. Meereboer et al. blended PHBV with cellulose acetate (CA) in the presence of a plasticizer, triethyl citrate (TEC), and chain extender poly(styrene-acrylic-co-glycidyl methacrylate) to enhance their ductile and impact properties. They reported complete immiscibility despite the similar Hansen solubility parameters, but improved impact strength by 110% compared to the virgin PHBV [11].

The aim of this study was to prepare PHBV thin films with bio-polyester PCL-diol in different ratios using an easy and cost-effective solution casting method, reducing energy consumption and avoiding thermal degradation during processing. PCL-diol was selected as a blending polymer with regard to its good miscibility with PHBV and its solubility in chloroform. Furthermore, there is no evidence in the available literature about polymer films based on PHBV and PCL-diol. A complete morphological, structural, thermal, mechanical, and surface characterization of the developed materials was conducted and evaluated at the same time to assess the most adequate formulation.

## 2. Materials and Methods

### 2.1. Materials

Poly(3-hidroxybutyrate-co-3-hydroxyvalerate) (PHBV) was manufactured by Tianan Biological Materials Co., Ltd. (Ningbo, China) and commercialized in pellet form under the grade name ENMAT Y 1000P. According to the manufacturer, PHBV has the following properties: M_w_~240,000 g/mol, density = 1.25 g/cm, *T*_g_ = 8 °C and *T*_m_ = 165 °C, hydroxyvalerate (HV) content 3 mol%. Poly(caprolactone diol) (M_n_~2000 g/mol) and chloroform (pro analysis) were provided by Sigma-Aldrich (St. Louis, MO, USA).

### 2.2. Preparation of PHBV/PCL-Diol Blends

PHBV pellets were dried before solution preparation at 50 °C for 5 h. Polymer blend solutions were prepared by mixing PHBV pellets and PCL-diol (40:60; 50:50; 60:40 wt% ratio) by dissolving in chloroform under continuous stirring for 6 h at elevated temperature (60 °C). PHBV/PCL-diol polymer blend solutions (prepared with 7 wt% of polymer in chloroform) were cast into a Petri dish covered with a glass lid and enabled to evaporate in a fume hood at ambient conditions.

### 2.3. Sample Characterization

The chemical structure of the neat polymers and the obtained samples was investigated with a Shimadzu IRaffinity-1s Fourier Transform Infrared Spectrometer (Kyoto, Japan) with Attenuated Total Reflectance (MIRacle 10 ATR-FTIR; Dia/ZnSe) in the wavelength range 4000 to 400 cm^−1^ by averaging 40 scans at a spectral resolution of 4 cm^−1^.

The surfaces and the microstructure of the prepared samples were examined with a scanning electron microscope (JEOL JSM-6460, Japan). Before the examination, all samples were vacuumed, gold-sputter coated, and observed at an accelerating voltage of 20 kV under different magnifications (1000 and 3000×). 

The wettability of the films was determined using the sessile drop method, using distilled water at room temperature (25 ± 1 °C). The contact angle of the tested liquid on the film surface was assessed using the Ossila Contact Angle software, version 1.3.0.0. (Sheffield, UK) To obtain accurate measurements, five readings were taken for each sample, and the reported value is the average of these measurements.

The surface characteristics of prepared PHBV/PVL-diol blends were analyzed via Atomic Force Microscopy (AFM). AFM evaluations were conducted using a CP-II device manufactured by Veeco in the Plainview, NY, USA. The measurements were carried out in a non-contact mode, employing a silicon-nitride probe with a symmetrically etched tip that had a radius of 10 ± 2 nm. Three different areas, namely 50 × 50 µm, 20 × 20 µm, and 10 × 10 μm, were scanned during the experiments with four repetitions on different locations on the samples. High-resolution images with dimensions of 256 × 256 pixels were acquired. During these measurements, specific parameters were set as follows: the probe’s vibration frequency was set at 270 kHz, the probe’s vibrating amplitude was maintained at 10 Å, and the scanning rate was set to 0.3 Hz. The setpoint for the measurements was established at 0.35 μm, with a gain of 0.25. Subsequently, all the data obtained from the AFM measurements were subjected to analysis using image analysis software called SPIP 6.2.0, which was developed by Image Metrology in Denmark. This software facilitated the calculation of various surface roughness parameters from the acquired AFM images. Average values of surface roughness parameters were calculated for each area of AFM analysis.

Thermal properties were evaluated by differential scanning calorimetry (DSC) on TA Instruments Q20 equipment (New Castle, DE, USA) under a nitrogen atmosphere (flow rate 50 cm^3^/min). The instrument’s temperature and cell constant calibration were performed using Indium reference samples, and C_p_ calibration was done with sapphire crystal, provided by TA Instruments. All of the samples were first cooled from room temperature to −90 °C followed by heating to 250 °C at a cooling and heating rate of 10 °C/min. Melting temperatures, *T_m_*, were taken as the peak temperatures, and glass transition temperatures, *T_g_*, were taken as the mid-point of the heat capacity change (∆Cp), both with standard uncertainty u(T) = 0.5 °C. The data were evaluated and presented using the TA Analyzer software package, version 11.0. The degree of crystallinity (*X_c_*) of the neat polymers and blends was calculated using the following Equation (1):(1)$$Xc=\frac{\Delta H_{m}}{\Delta H_{ref}}\;\cdot 100\%$$
where Δ*H_m_* is a measured enthalpy change during the melting determined from the DSC curve and Δ*H_ref_* is the theoretical enthalpy of melting for a fully crystalline polymer and blend. For blends, Δ*H_ref_* is calculated using the following Equation (2):(2)$$∆H_{ref\;}=\omega 1\cdot \Delta H_{m1}+\omega 2\cdot \Delta H_{m2}$$
where ω is a weight fraction of individual polymers. Melting enthalpy of 100% crystalline PHBV and PCL-diol have been reported to be 146 J/g and 135 g/J, respectively [23,24].

Tensile strength and elongation at the break of the prepared films were measured using the Universal Testing Machine Shimadzu EZ-LX test machine (Shimadzu, Kyoto, Japan) according to the guidelines of ASTM Standard ASTM D882-18 [25]. Samples were previously shaped into rectangular strips and stretched with a cross-head speed set at 10 mm/min. Measurements were repeated five times for each sample, and tensile strength (TS) and elongation at break (EB) were estimated at the average value. 

## 3. Results and Discussion

### 3.1. Appearance of PHBV/PCL-Diol Films

Prepared blends under observation have unveiled a striking divergence in appearance and texture related to their composition, i.e., the PHVB and PCL-diol ratios in the blend composition (Figure 1). In Figure 1a, it was observed that neat PHBV was unable to form a continuous film covering the bottom of the Petri plate effectively. However, a significant improvement was achieved when PCL-diol was added to the blend composition at a concentration of 40 wt% (Figure 1b). When the amount of PCL-diol in the blend was increased to 50 and 60 wt%, as shown in Figure 1c and Figure 1d, respectively, continuous films were successfully formed. This indicates that the addition of PCL-diol significantly enhances the film-forming ability of PHBV. This improvement can likely be attributed to the arrangement of PCL-diol molecules between the PHBV molecules, which facilitates the formation of a continuous and cohesive film.

### 3.2. Structural Analysis

The FTIR spectra of neat biopolymers and blends are presented in Figure 2. In the IR spectrum of neat PCL-diol (Figure 2a), there is a peak with the center at 3554 cm^−1^ that corresponds to OH stretching. Two peaks at 2942 and 2864 cm^−1^ correspond to CH stretching. In the FTIR spectra of PHBV (Figure 2b) and blends in this range, there are three peaks (from CH_2_ and CH_3_). An intense peak at 1721 cm^−1^ is associated with the CO stretching vibration of ester bonds. Bands at 1473, 1420, and 1393 cm^−1^ are attributed to CH_2_ deformation, OH bending, and COH bending in PCL-diol (Figure 2a). In the FTIR spectra of PHBV and blends, two bands at 1459–1455 and 1379 cm^−1^ are assigned to CH_2_ deformation. Peaks at 1293, 1240, 1172, and 1047 cm^−1^ are attributed to COC stretching in PCL-diol. In the spectra of PHBV and blends OC stretching vibrations manifest as double peaks at 1279 and 1260 cm^−1^, as well as bands at 1180–1184 cm^−1^, and double peaks at 1053 and 1045 cm^−2^. FTIR spectra of prepared blends closely resemble the spectrum of neat PHBV in terms of peak positions and intensities, reporting phase compatibility implying that the blend behaves as a single-phase material with no interactions between the compounding biopolymers.

### 3.3. Morphological Analysis

SEM micrographs of a neat PHBV and blends are presented in Figure 3. The micrograph of neat PHBV (Figure 3a) shows that it has difficulty forming a continuous film. This is due to the inherent properties of PHBV. PHBV is not soluble in most solvents except chloroform. However, the micrograph suggests that this film is not uniform and has voids, indicating that the PHBV chains in the film are not well-extended and are more coiled up on themselves. This conformation can create voids between the polymer chains within the film. PCL-diol is a more flexible biopolymer compared to PHBV. The addition of PCL-diol improves the film-forming properties of PHBV by penetrating and filling the voids or gaps between the coiled PHBV chains. This leads to a more densely packed and continuous film structure. The micrographs indicate that the optimal blend ratio of PCL-diol is 60 wt%. At this ratio, the PCL-diol effectively fills the voids, resulting in a smoother and more uniform film. This improved film appearance is likely due to the better intermolecular interactions and compatibility between PHBV and PCL-diol.

### 3.4. Static Water Contact Angles

Contact angle determination is important for various applications because it provides valuable information about the wettability of a material, which in turn helps to understand how the material interacts with liquids. Figure 4 shows the contact angle values for neat PHBV and blends. It is evident from the data that the contact angle values vary with the composition of the blends. The highest contact angle observed for neat PHBV (70.8°) indicates that it has a moderately hydrophilic surface. With the increase in PCL-diol in the blend structure, the contact angle decreases. This suggests that the introduction of PCL-diol into the blend increases the surface energy and wettability of the material. The contact angle approaching that of neat PCL-diol (57.03°) [26] indicates that the surface properties of the blends are becoming more similar to those of PCL-diol. Therefore, the lowest value was recorded for the sample PHBV/PCL-diol 60/40 (58.62°). The decrease in contact angle with increasing PCL-diol content can be explained by the fact that PCL-diol is likely more hydrophilic (water-attracting) than PHBV. When analyzing the results, the influence of film porosity on the value of the contact angle was also taken into account; however, the same angle values were recorded at the moment of dropping the drop and after 90 s, so this factor is excluded from the consideration.

### 3.5. Atomic Force Microscopy (AFM)

Figure 5 shows the 3D topography images of different blends. Measurements on surfaces of 50 × 50 µm can be considered the domain of microtopography—microroughness—while measurements on surfaces of 10 × 10 µm belong to the domain of nanotopography—nanoroughness. From Figure 5, it can be concluded that samples with the lowest amount of PCL-diol (40 wt%) possess the roughest surface. Surface roughness parameters of each blend are given in Table 1: arithmetical mean height (Sa), root mean square roughness (Sq), skewness (Ssk), kurtosis (Sku), reduced peak height (Spk), and reduced valley depth (Svk), which were obtained for different areas (50 × 50 µm, 20 × 20 µm, and 10 × 10 µm). Examined surfaces do not have statistically significant differences in terms of the roughness parameter Sa. The microroughness of the surface of sample PHBV/PCL-diol 40/60 is the highest, but on the other hand, the nanoroughness is the lowest. It could be explained by the fact that biopolymers stack effectively at the molecular level, creating smooth and uniform surfaces at the nanoscale. Sq value, as a quantitative measure of surface roughness, possesses the highest value for the sample PHBV/PCL-diol 40/60 at the microscale, but the lowest at the nanoscale. The reason lies in the fact that microscale observations may capture larger surface irregularities and variations, leading to a higher Sq value, which is especially evident in the formulation of blends where one component is in excess (40/60, 60/40). Therefore, the sample with the same amounts of biopolymers in the blend (50/50) possesses the lowest Sq values at the microscale. In contrast, nanoscale observations focus on much smaller details and may not pick up the same level of roughness. As the Ssk parameter indicates, the PHBV/PCL-diol 40/60 has the most symmetrical topography, without pronounced polarity, while two other samples have slightly pronounced negative polarity, i.e., surfaces with more depressions than protrusions. For all the tested samples, the Sku value was found to be very close to the number 3, which means that most of the surfaces are quite random and without any large occasional surface deviations in the form of individual high peaks and/or deep depressions.

### 3.6. DSC Analysis

The thermal characterization of the prepared samples was carried out using DSC (Figure 6). Obtained blends (Figure 6c–e) possess *Tg* values between the *Tg* values of neat biopolymers PHBV and PCL-diol. *Tg* values indicate that the blends have intermediate thermal characteristics compared to the neat biopolymers. In the thermograms of blends, there are two melting peaks (one between 48 and 50 °C, and the other one at 171–173 °C). The first peak corresponds to the melting of PCL-diol, and the second one to the melting of PHBV in the blend composition. The presence of two distinct melting peaks indicates that both PCL-diol and PHBV maintain their crystalline structures in the blend. The fact that the peaks are at different temperatures suggests that these components do not form a single, homogenous crystalline phase but rather retain their distinct properties in the blend. However, the melting enthalpy values are lower for the blends than for the neat biopolymers, indicating that the crystalline regions of the polymers in the blend are less well-defined or less ordered compared to the neat polymers. The lower melting enthalpy values suggest that the degree of crystallinity in the blends is lower compared to the neat polymers. This could be due to interference in the crystallization process when the two polymers are mixed. The thermal properties of neat polymers and their blends were obtained from the DSC curves such as *Tg*, *Tm*, melting enthalpy (Δ*H_m_*), and degree of crystallinity (*X*_c_) calculated using Equations (1) and (2). The degree of crystallinity in polymer samples exceeds 30%, indicating a relatively high level of ordered molecular structure within the material (Table 2). 

### 3.7. Tensile Properties

Tensile strength (TS) and elongation at break (EB) values along with corresponding standard deviations are shown in Table 3 A neat PHBV has the greatest value of TS and ES. The addition of PCL-diol in the amounts of 40 and 50 wt% leads to a significant decrease in TS and EB values (up to 69.7 and 71.7%, respectively). The addition of PCL-diol in the amount of 60 wt% results in an increase in TS value of 55.2%, while EB values slightly decrease (relative to the sample PHBV/PCL-diol 50/50). Tensile strength and elongation at break are inversely related. Strong intermolecular forces lead to higher tensile strength because the forces required to break these bonds are greater. On the other hand, strong forces limit the material’s ability to stretch significantly before failure, leading to a lower EB. These results indicate that in this specific composition, the PCL-diol is somehow enhancing the tensile strength of the blend, possibly due to specific interactions or compatibility between PHBV and PCL-diol at this ratio. However, it is noted that the elongation at break values slightly decreases which suggests that the material becomes slightly less stretchable with further addition of PCL-diol (above 50 wt%). The tensile stress-strain diagram is in the Appendix A.

## 4. Conclusions

The study found that the blending of PHBV with PCL-diol using a solution casting method results in an improvement in PHBV film-forming ability and morphological properties. SEM analysis has shown that the sample with the greatest amount of PCL-diol (60 wt%) forms a smoother and more uniform film because the PCL-diol effectively fills the voids. Results of AFM analysis reveal that the microroughness of the surface of sample PHBV/PCL-diol 40/60 is the highest, but on the other hand, the nanoroughness is the lowest. It could be explained by the fact that biopolymers stack effectively at the molecular level creating smooth and uniform surfaces at the nanoscale. Contact angle measurements have shown that the introduction of PCL-diol into the blend increases the surface energy and wettability of the material. These results suggest that the PHBV/PCL-diol blend with optimal composition (40/60) meets the required performance standards and environmental objectives and could be used in agriculture (seed encapsulation) and for biomedical applications (wound dressings and drug delivery systems).

## Figures and Tables

**Figure 1 polymers-15-04694-f001:**
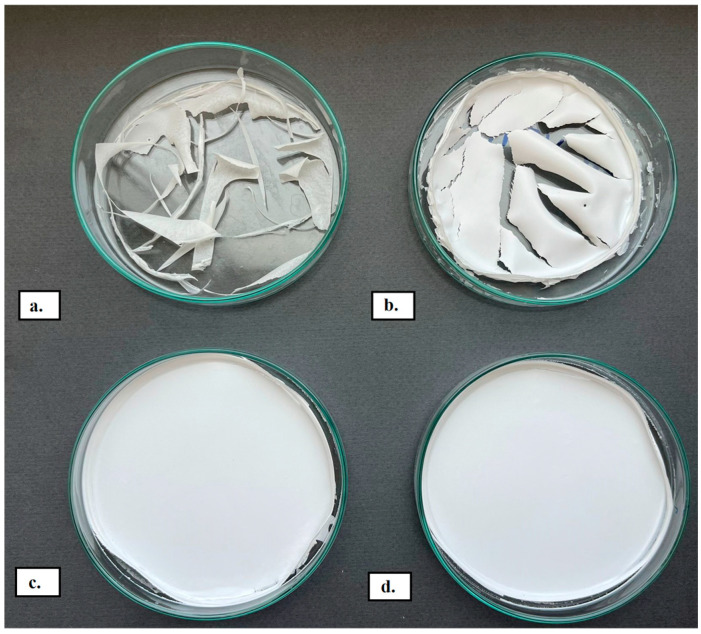
The appearance of (**a**) neat PHBV and its blends with PCL-diol: (**b**) PHBV/PCL-diol 40/60; (**c**) 50/50; (**d**) 60/40.

**Figure 2 polymers-15-04694-f002:**
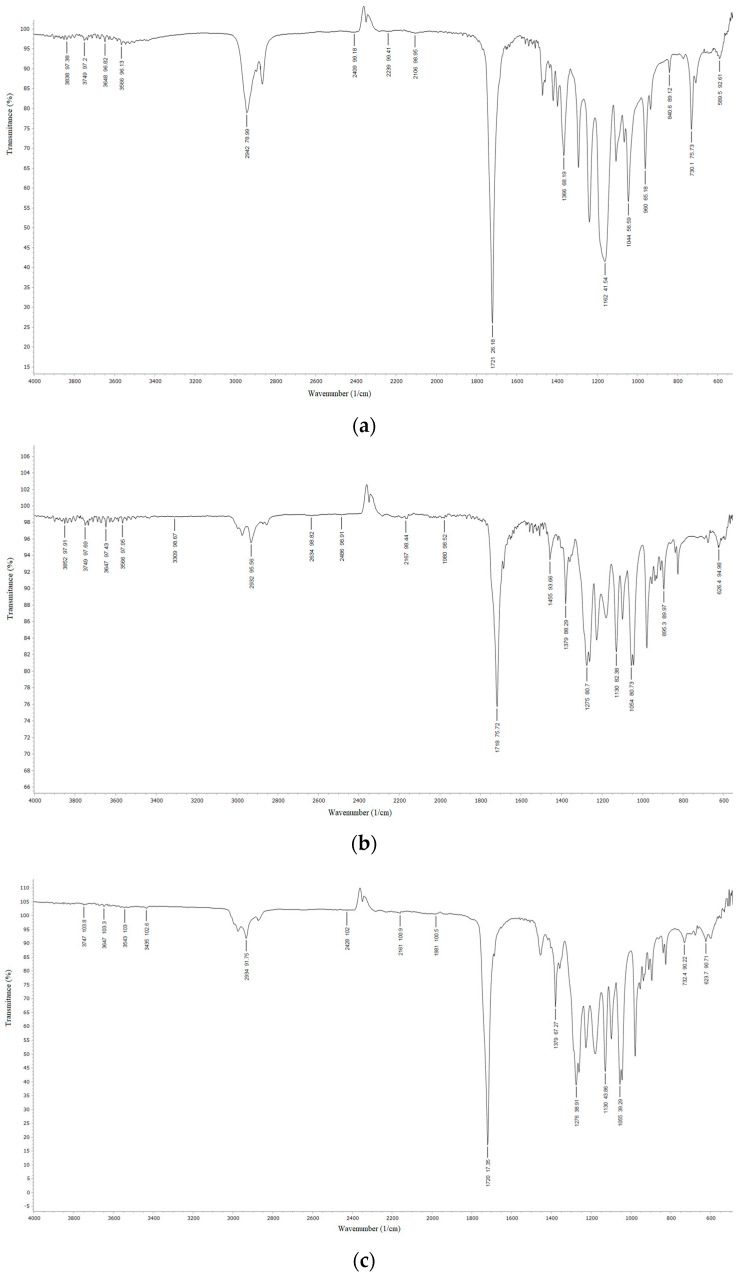
FTIR spectra of (**a**) PCL-diol, (**b**) PHVB film, (**c**) PHVB/PCL-diol 60/40, (**d**) PHVB/PCL-diol 50/50, and (**e**) PHVB/PCL-diol 40/60.

**Figure 3 polymers-15-04694-f003:**
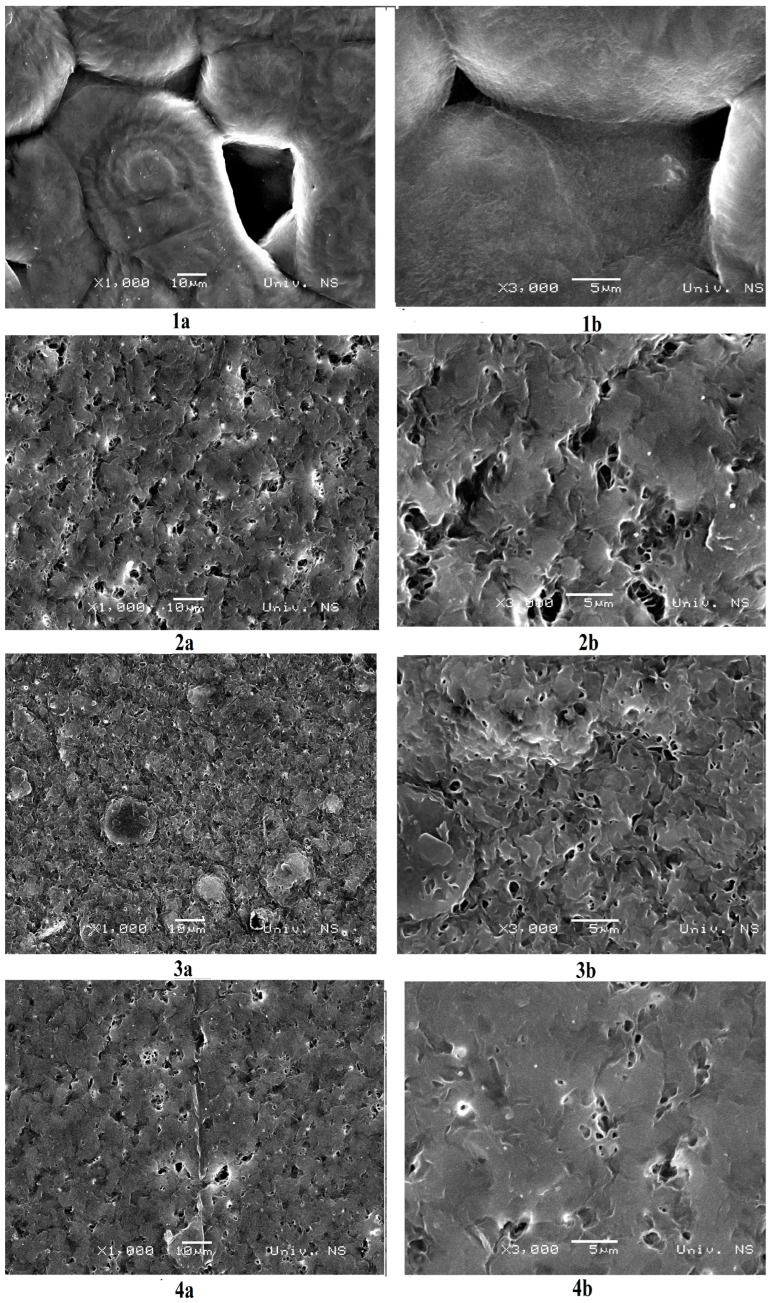
SEM micrographs at magnifications of (**a**) 1000 and (**b**) 3000×: 1. neat PHBV film; 2. PHBV/PCL-diol 60/40 film; 3. PHBV/PCL-diol 50/50 film; 4. PHBV/PCL-diol 40/60 film.

**Figure 4 polymers-15-04694-f004:**
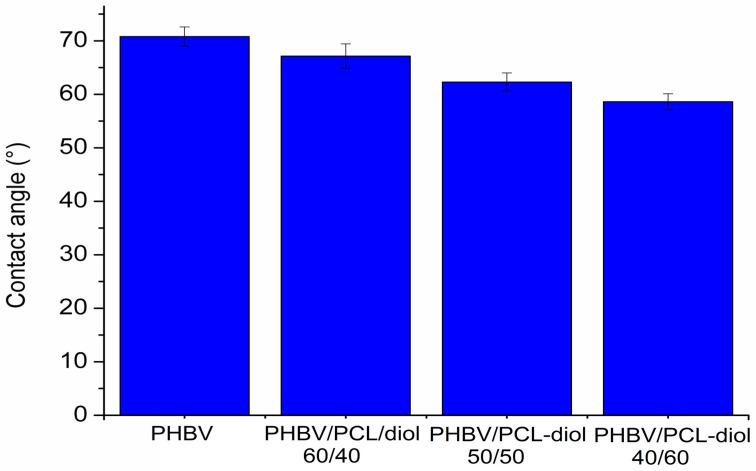
Contact angle measurements of neat PHBV–based film and PHBV/PCL-diol blends.

**Figure 5 polymers-15-04694-f005:**
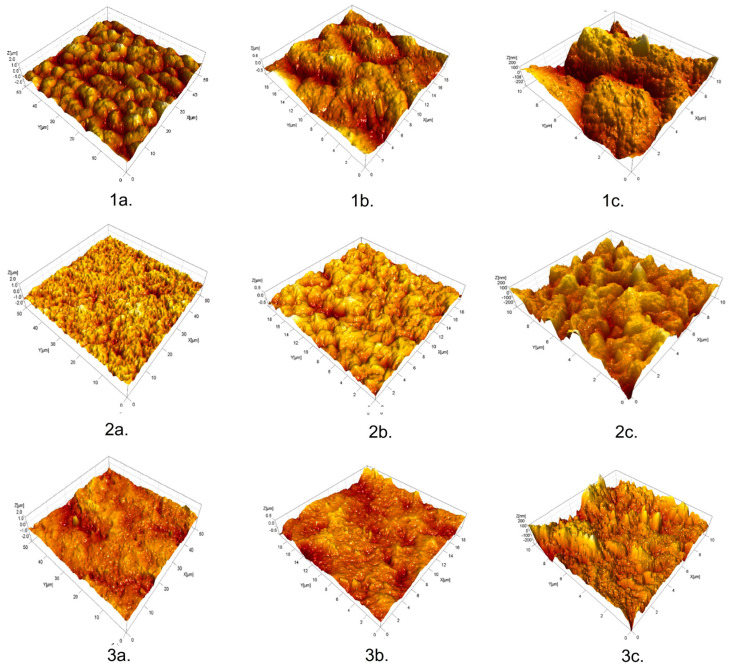
Three-dimensional atomic force microscopy (AFM) images ((**a**) 50 × 50 µm, (**b**) 20 × 20 µm, (**c**) 10 × 10 µm) of the PHBV/PCL-diol blends: 1. 40/60; 2. 50/50; 3. 60/40.

**Figure 6 polymers-15-04694-f006:**
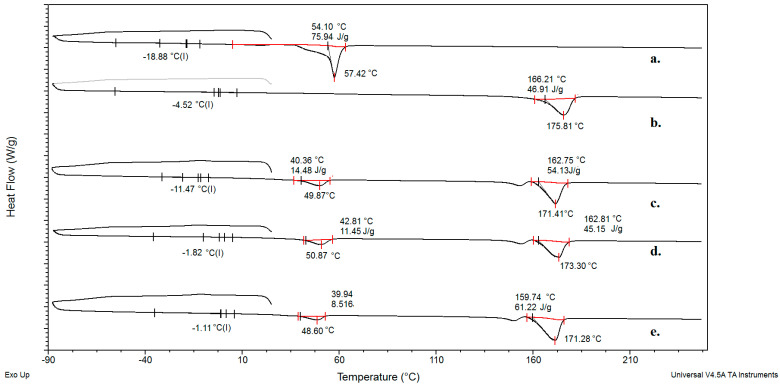
DSC thermogram of (**a**) PCL-diol, (**b**) PHBV, (**c**) PHBV/PCL-diol 60/40, (**d**) PHBV/PCL-diol 50/50, and (**e**) PHBV/PCL-diol 40/60.

**Table 1 polymers-15-04694-t001:** Surface roughness parameters of blends determined from AFM measurements.

Sample	Scanned Area (µm)	Sa (nm)	Sq (nm)	Ssk	Sku	Spk (nm)	Svk (nm)
PHBV/PCL-diol 40/60	50 × 50 20 × 20 10 × 10	161.4 ± 19.8	202.7 ± 21.2	0.03 ± 0.11	3.03 ± 0.24	208.9 ± 13.8	183.9 ±14.2
		122.6 ± 21.6	152.7 ± 20.3	−0.07 ± 0.23	2.95 ± 0.22	141.6 ± 12.2	144.3 ±18.4
		100.3 ± 15.6	124.9 ± 21.6	−0.10 ± 0.32	2.92 ± 0.32	115.4 ± 7.1	120.0 ± 12.65
PHBV/PCL-diol 50/50		128.8 ± 7.6	161.7 ± 9.1	−0.37 ± 0.13	3.18 ± 0.11	135.3 ± 5.7	181.5 ± 12.9
		106.9 ± 8.5	135.7 ± 9.6	−0.40 ± 0.13	3.52 ± 0.12	112.8 ± 20.6	164.9 ± 15.1
		85.1 ± 12.4	106.5 ± 15.2	−0.43 ± 0.11	3.22 ± 0.14	77.3 ± 6.8	124.9 ± 18.0
PHBV/PCL-diol 60/40		186.2 ± 16.8	238.8 ± 18.4	−0.53 ± 0.23	3.68 ± 0.22	162.3 ± 15.6	320.3 ± 24.3
		152.1 ± 12.13	196.8 ± 25.1	−0.60 ± 0.13	3.77 ± 0.13	140.4 ± 14.3	276.6 ±24.2
		77.6 ± 13.2	98.7 ± 16.6	−0.25 ± 0.22	3.69 ± 0.21	90.7 ± 14.8	115.6 ± 14.5

**Table 2 polymers-15-04694-t002:** Thermal properties of neat PCL-diol, PHBV, and PHBV/PCL-diol blends obtained from DSC.

Sample	*Tg*, °C	*Tm*, °C	Δ*H_m_*, J/g	*X*_c_, %
PCL-diol	−18	57	75	55.5
PHBV	−4	175	46	31.5
PHBV/PCL-diol 40/60	−11	49 171	14 54	48.7
PHBV/PCL-diol 50/50	−1	50 171	11 45	46.9
PHBV/PCL-diol 60/40	−1	48 171	8 61	48.7

**Table 3 polymers-15-04694-t003:** Tensile strength (TS) and elongation of a break for neat PHBV biopolymer and PHBV/PCL-diol blends.

Sample	TS (N/mm^2^)	EB (%)
Neat PHBV	11.07 ± 4.38	5.69 ± 2.15
PHBV/PCL-diol 60/40	4.80 ± 1.32	1.61 ± 0.74
PHBV/PCL-diol 50/50	3.35 ± 0.65	2.37 ± 0.86
PHBV/PCL-diol 40/60	5.20 ± 2.85	2.03 ± 1.86

## Data Availability

Data are contained within the article and Appendix A.

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
