# Peer review of "Preparation and Characterization of PHBV/PCL-Diol Blend Films"

_polymers, 2023, doi:10.3390/polym15244694_

Round 1
Reviewer 1 Report
Comments and Suggestions for Authors
The authors have presented interesting research entitle "Preparation and characterization of PHBV/PCL-diol blend films". However the major concern is that only fabrication and characterisation of polymer blend in any form does not make the research investigated a proof of concept. Thus I suggest for addition of potential application in term of packaging or something else relevant to polymer blend followed with correction as minor comments in manuscript.
Suggested to indicate the wave number in the figure 2, which will make readability easily.
Suggested to add error bars in figure 4, since the experiments are performed in triplicate definitely there will be error bars with application of stats and comparison of neat film contact angle data with others.
Supplement the shear stress graph for tensile strength graph.
Thanks and good luck
Reviewer 2 Report
Comments and Suggestions for Authors
The manuscript presents interesting and useful data about the preparation and characterization of solution-casted PHBV/PCL-diol films. A morphological, structural, thermal, mechanical, and surface characterization of the developed materials was conducted. Blending is a simple and effective approach for obtaining new polymeric materials with improved properties. In addition, biodegradable blends employ conventional technology at low cost and have high potential applications. However, some issues need to be addressed. Hence, I recommend this manuscript be published in Polymers after the author addresses the following concerns:
1. In the introduction the authors claim that “In general, there is a small number of publications devoted to the development and modification of PHBV“, while in fact, it is one of the most widely studied microbially synthesized copolymers. Please take a look at some recent review articles such as Lhamo and Mahanty, J Polym Environ 31, 4641–4661 (2023) or Rivera-Briso, Serrano-Aroca, Polymers 10, 732 (2018) and references therein. Some more recent references should be cited in the text.
2. As the properties of the PHA polymers are clearly dependent on the chain length, The Mw of the used PHBV and the PHV content should be given in the text.
3. FTIR analysis – There is no description for Fig 2e – I guess it is neat PCL-diol? The authors claim that “In the IR spectrum of neat PCL-diol, there is a peak with the center at 3554 cm-1” – I do not see the peak, if it is there, it should be marked, or better that part of the graph should enlarge for clarity.
Line 186: “Two peaks at 2942 and 2864 cm-1 correspond to CH stretching (from CH2 )” – is it from CH or CH2?
Line 190: “C-O-H bending in PCL-diol (Figure 2a)” – Fig 2a is denoted as PHVB
Line 191: “Peaks at 1293, 1240, 1172, and 1047 cm−1 are attributed to O–C stretching in ester groups in PCL-diol” – these are rather contributions from C-O-C ether group
What do the FTIR results show about the blend properties? Some discussion should be provided.
4. Static water contact angles – in line 225 it is written “The highest contact angle observed for neat PHBV (70.8 °) indicates that it has a relatively hydrophobic surface”. Generally, if the water contact angle is smaller than 90°, the solid surface is considered hydrophilic. The term “moderately hydrophilic” would be more appropriate.
5. AFM
Line 249: “The microroughness of the surface of sample PHBV/PCL-diol 40/60 is the highest, but on the other hand, the nanoroughness is the lowest.” The statement does not agree with the values in Table 1. The same issue while discussing Sq values (line 253). Are the values in Table 1 given inversely for PHBV/PCL-diol 60/40 and 40/60? As the authors noticed, the roughness parameters may vary on the same sample surface, depending on the exact spot of the measurement. To address this issue the calculations should be repeated for several areas of different spots and the average value should be calculated. Also, uncertainties of the measurement should be provided. In Table 1, it should be specified which row corresponds to each resolution. Moreover, the z-scale in Figure 5 is illegible.
6. DSC
The Tg values in the Fig.6 are not marked well. Too many ticks are confusing. The Tg region should be enlarged. It seems that the Tg values of blends are higher than those of both neat polymers. How would the authors explain this? The formation of double melting peaks of PHBV should be discussed. The crystallinity of the material is an important factor that has an influence on the mechanical and optical properties as well as the rate of biodegradation of materials. The degree of crystallinity of the neat materials and the obtained blends should be provided -it could be calculated from DSC measurements. The XRD /WAXS experiments could also be beneficial to get insight into the crystallization process of the blends. Could the authors provide them?
7. Tensile properties
It was shown that the addition of PCL-diol results in a decrease in TS, but also a decrease in elongation at break compared to neat PHVB. Usually, materials with a lower tensile strength have a higher elongation at break values. Such a trend was also observed for PHVB/PCl blends in the literature, e.g. in ref 12. How would the authors explain this? Could the TS value of neat PCl diol be added to get insight into the interactions between the two polymers?
Minor comments:
p.1. line 41– Bacillus is a Gram-positive bacteria, not Gram-negative
p.3. line 106 - Poly(3-hidroxybutyrat-co-3-valerate)- should be hydroxybutyrate
Figure 3 – there is an inconsistency between the caption and picture descriptions, pictures are signed as 1a, 1b, 2a, 2b…, while in the caption there are only letters a b c d.
Comments on the Quality of English LanguageMinor editing of the English language required
Round 2
Reviewer 1 Report
Comments and Suggestions for Authors
The authors have significantly improve the present form of the manuscript after implementing all the suggestion with their hard work. Therefore, I recommend the manuscript for further processing.
Author Response
Thank you very much for your recommendation.
Reviewer 2 Report
Comments and Suggestions for Authors
The authors have paid attention to my previous critiques. In answer, they have revised and improved the manuscript. Thus, it is now more presentable for publication.
Minor comments:
p. 2, line 85 - the second bracket is missing
p. 3, line 123 – it should be Poly(3-hydroxybutyrate-co-3-valerate)
p. 5, paragraph 3.2 – the information about the additional FTIR spectra in the Supplementary should be included in this paragraph
Table 1 – in the case of Ssk, the number of significant digits should be reconsidered, e.g. the value of 0.03 with 0.1 error doesn’t look good
Comments on the Quality of English Language
There are some formatting errors in the article
